# Turn-key constrained parameter space exploration for particle accelerators using Bayesian active learning

Ryan Roussel [1✉], Juan Pablo Gonzalez-Aguilera [1], Young-Kee Kim [1], Eric Wisniewski[2], Wanming Liu[2], Philippe Piot[2,3], John Power[2], Adi Hanuka[4] & Auralee Edelen[4]

Particle accelerators are invaluable discovery engines in the chemical, biological and physical sciences. Characterization of the accelerated beam response to accelerator input parameters is often the first step when conducting accelerator-based experiments. Currently used techniques for characterization, such as grid-like parameter sampling scans, become impractical when extended to higher dimensional input spaces, when complicated measurement constraints are present, or prior information known about the beam response is scarce. Here in this work, we describe an adaptation of the popular Bayesian optimization algorithm, which enables a turn-key exploration of input parameter spaces. Our algorithm replaces the need for parameter scans while minimizing prior information needed about the measurement's behavior and associated measurement constraints. We experimentally demonstrate that our algorithm autonomously conducts an adaptive, multi-parameter exploration of input parameter space, potentially orders of magnitude faster than conventional grid-like parameter scans, while making highly constrained, single-shot beam phase-space measurements and accounts for costs associated with changing input parameters. In addition to applications in accelerator-based scientific experiments, this algorithm addresses challenges shared by many scientific disciplines, and is thus applicable to autonomously conducting experiments over a broad range of research topics.

[1] Department of Physics, University of Chicago, Chicago, IL 60637, USA. [2] Argonne Wakefield Accelerator, Argonne National Laboratory, Lemont, IL 60439, USA. [3] Department of Physics, Northern Illinois University, DeKalb, Illinois 60115, USA. [4] SLAC National Laboratory, Menlo Park, CA 94025, USA. ✉email: rroussel@uchicago.edu

Particle accelerators have enabled ground-breaking discoveries in the fields of chemistry[1], biology[2], and physics[3,4]. They are also increasingly deployed for societal applications, such as in medical[5] or industrial[6] fields. During operation, accelerator parameters need to be tuned to produce beams with specific characteristics that match the needs for front-end applications. Measuring these beam properties as a function of one or more input parameters using limited diagnostics and time-consuming measurements is a necessary part of operations, experimental planning, and tolerance determination. This comes at the expense of reducing accelerator availability for experimenters. These challenges are shared by many different scientific fields, which try to characterize complex, highly nonlinear and correlated systems, using difficult to execute scientific measurements and complicated diagnostics[7,8].

Due to the complex and time-consuming nature of accelerator measurements, characterization of the beam response to input parameters is often limited to simple, uniformly spaced, grid-like parameter scans in one or two dimensions. This limitation results from the poor scaling of grid-like scans to higher-dimensional spaces, where the number of samples grows exponentially with the number of input parameters. Furthermore, despite its simplicity at face value, it is often difficult to determine the ideal properties of parametric scans which will result in successful and efficient sampling. A predefined grid spacing ultimately limits the ability to resolve fine features while potentially oversampling slow variations of the measured parameters. As a result, specifying the scan parameters a priori requires prior information about the measurement's functional dependence on each parameter. This slows down frequent routine studies and makes characterization of novel measurements difficult to execute successfully.

The existence of tight constraints in input space which determine if measurements are viable further complicates this process. Upper and lower input parameter limits are often determined by practical constraints of conducting measurements. For example, transverse beam size measurements on diagnostic screens are limited by the screen size (available field of view), which in turn, imposes limits on the strength of upstream focusing magnet parameters. Simulation studies or extra measurements are needed beforehand to determine these limits. Even when simulations are available, they do not necessarily represent realistic machine behavior. Measurements, on the other hand, may become inaccurate due to time dependent changes in the accelerator, further complicating this problem. Furthermore, while these limits can be easily determined for a single parameter experimentally, it becomes practically infeasible to efficiently determine limits in higher-dimensional input spaces, as they are often correlated with multiple parameters. Limitations such as these are shared among many types of scientific experiments[9].

Finally, it is desirable to prevent rapid changes in accelerator input parameters during operation. In some cases, it is temporally expensive to make changes in parameters, such as when mechanical actuators are used to change the phase of accelerating cavities. In other cases, fast feedback algorithms used in accelerator subsystems rely on adiabatic changes in external parameters to maintain system stability. Large jumps in parameter space can delay convergence of these feedback systems or worse, cause them to fail entirely. Practical experimental considerations such as these must be taken into account when automated sampling algorithms are used, striking a balance between costs associated with changing input parameters and the expected information gained by candidate samples.

In this work, we construct an algorithm that replaces simple parametric scans for characterizing a target function of interest, while meeting poorly understood requirements for practical measurements in a flexible manner. The algorithm is "turn-key", requiring as little prior information about the target function and measurement constraints as possible. As a result, our algorithm reduces beamline tuning time during normal operations, while enabling efficient exploration of novel or poorly characterized systems. Our algorithm, coined "Constrained Proximal Bayesian Exploration" (CPBE), starts from an initial valid observation and sequentially samples points in input space that maximize information gain of the target function at each observation step. It dynamically adapts its sampling strategy to measured functional behavior with respect to each parameter and respects necessary constraints for successful measurements. Finally, the algorithm is biased towards making small jumps in input space, balancing the trade-off between exploration and costs associated with changing input parameters. While our focus here is the application of this method towards parametric accelerator exploration, this algorithm can be applied in any experimental scenario that requires parametric scans.

## Results

Our algorithm is an adaptation of Bayesian optimization (BO)[10,11], applied to maximizing information gain[12], which is often referred to as active learning or uncertainty sampling[13–16]. Bayesian optimization generally consists of two components. The first component of BO is a probabilistic surrogate model that predicts the probability distribution of the value $f$ as a function of the input parameter vector $\mathbf{x}$. Gaussian Processes (GPs)[17] are a popular model choice for Bayesian optimization. The behavior of a GP model is determined by a kernel function, which describes the correlation between function values based on their location in input space relative to previous measurements. The kernel function itself can depend on a collection of hyperparameters, which often includes a length scale hyperparameter. This parameter describes the effective smoothness of the model, where small and large values correspond to rapidly and slowly varying functional behavior respectively. An independent length scale hyperparameter can be specified for each input parameter, in a process known as automatic relevance determination (ARD)[18]. These hyperparameters are determined by maximizing the marginal log likelihood of the GP model[17], conditioned on experimental measurements, which balances model accuracy and complexity.

The second component of BO is an acquisition function, which characterizes the value gained by observing a particular point in input space. Bayesian optimization selects the next experimental sample by finding the point in input space which maximizes the acquisition function. Several different acquisition functions that are commonly used for optimization are Probability of Improvement[19], Expected Improvement[20], and Upper Confidence Bound (UCB)[12]. The UCB acquisition function is defined as

$$\alpha(\mathbf{x}) = \mu(\mathbf{x}) + \sqrt{\beta}\sigma(\mathbf{x}) \qquad (1)$$

where $\mu(\mathbf{x})$ is the predicted mean of the function value and $\sigma(\mathbf{x})$ is the predicted uncertainty, both determined by the GP model. This acquisition function is of particular interest as it allows the user to specify the optimization parameter $\beta$, which represents the trade-off between exploitation (sampling to take advantage of predicted extrema) and exploration (sampling to reduce prediction uncertainty). An optimizer with a small value of $\beta$ prefers exploitation, while a large value of $\beta$ prioritizes exploration. It has been shown that maximum information gain of the GP model occurs when the acquisition function is only comprised of the uncertainty factor $\alpha(\mathbf{x}) = \sigma(\mathbf{x})$[12], effectively choosing $\beta \to \infty$. The uncertainty predicted by the GP model is dependent only on the locations of previously sampled points in input space (see Eq. (7) in Methods Section for details).

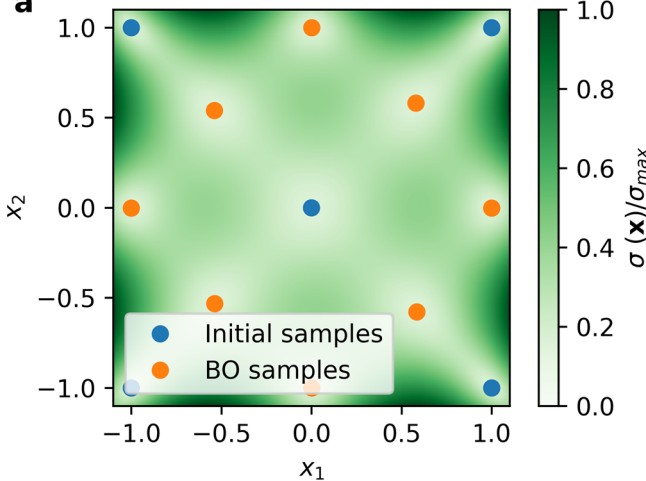

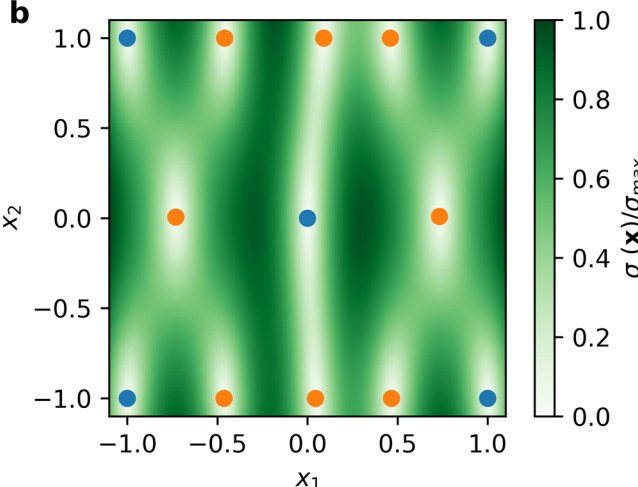

**Fig. 1 Plots showing Bayesian optimization (BO) sampling patterns depending on radial basis function (RBF) kernel length scales and using $\alpha(\mathbf{x}) = \sigma(\mathbf{x})$ as the acquisition function.** Blue points are initial samples and orange points are sampled during Bayesian optimization. **a** Length scales for both $[x_1, x_2]$ set to 1. **b** Length scales for variables $[x_1, x_2]$ set to [0.25, 1].

Figure 1 shows the effect of kernel length scales on Bayesian optimization when we use this exploratory acquisition function. If the GP kernel has a single length scale, then the acquisition function is maximized at points that are the largest distance away from all previous observations. The sampling algorithm will choose points that form a grid-like pattern in input space, dependent on where initial samples were taken. However, if the length scale is different along each input axis, the exploration algorithm will be biased towards sampling points along the axis associated with the shortest length scale. This leads to an efficient sampling strategy, as more samples are needed to resolve rapidly changing features of the function, while fewer samples are used when the function changes slowly. We use this behavior as a starting point for creating the CPBE acquisition function.

We define the CPBE acquisition function to be

$$\alpha(\mathbf{x}, \mathbf{x_0}) = \sigma(\mathbf{x})\Psi(\mathbf{x}, \mathbf{x_0})\prod_{i=1}^{N} P_i[g_i(\mathbf{x}) \geq h_i] \quad (2)$$

$$\Psi(\mathbf{x}, \mathbf{x_0}) = \exp\left(-\frac{1}{2}(\mathbf{x}-\mathbf{x_0})^T \mathbf{\Sigma}^{-1}(\mathbf{x}-\mathbf{x_0})\right) \quad (3)$$

where the two additional terms have the following effects.

The factor $\Psi(\mathbf{x}, \mathbf{x_0})$ in Eq. (2) represents a proximal biasing factor where $\mathbf{x_0}$ is the most recently sampled point in input space. The covariance matrix $\mathbf{\Sigma}$ specifies the length scale at which points are biased, where a smaller element in the matrix leads to a stronger biasing towards the most recently observed point. By including the proximal term, we bias the acquisition function away from points that are not within a close proximity to the most recently observed point, while still allowing exploration in cases where large jumps in input space result in highly valued observations. This factor is included for two reasons. First, changing parameters in a real accelerator generally incurs a temporal cost, proportional to the change in the parameter. Second, changing accelerator parameters quickly can disrupt rapid feedback systems used to maintain supporting accelerator subsystems (such as those which stabilize the phase and amplitude of radio-frequency fields in accelerating cavities).

The last factor, $\prod_{i=1}^{N} P_i[g_i(\mathbf{x}) \geq h_i]$ represents the multiplicative probability that $N$ operational constraints are satisfied, following[21]. This factor weights the CPBE acquisition function by the probability that a constraining function $g_i(\mathbf{x})$ is greater than a predetermined scalar value $h_i$. As a result, this factor will bias our acquisition function against sampling in regions of input space that are not likely to satisfy the constraints. The probability is calculated based on GP model predictions of $\mu_i(\mathbf{x})$ and $\sigma_i(\mathbf{x})$ of the individual constraining functions $g_i(\mathbf{x})$, giving

$$P_i[g_i(\mathbf{x}) \geq h_i] = 1 - \Phi\left(\frac{h_i - \mu_i(\mathbf{x})}{\sigma_i(\mathbf{x})}\right) \quad (4)$$

where $\Phi(x)$ is the Gaussian cumulative distribution function.

**Experimental demonstration**. We conducted an experiment at the Argonne Wakefield Accelerator (AWA) to demonstrate the application of Bayesian exploration to enable beamline characterization with a single-shot emittance measurement. The AWA beamline accelerates electrons to ~42 MeV using a photoinjector electron source, combined with a normal-conducting linear accelerator[22]. The transverse phase-space area or "emittance" of beams produced by the accelerator is an important figure of merit that must be minimized to ensure ideal transport of the beam through the accelerator and meet specific experimental criteria. The emittance ultimately sets the beam brightness, a critical parameter in accelerator-based light sources[23] and colliders[24]. The emittance is sensitive to several beamline parameters summarized in Fig. 2, including the magnetic field strength of a pair of solenoidal lenses surrounding the photoinjector (referred to here as the "focusing" solenoid and characterized by the scaling parameter K0) and a solenoid in between the photoinjector and the first accelerating cavity (referred to here as the "matching solenoid" and controlled via the scaling parameter K1). Our goal is to explore the emittance response to these solenoids, as well as two quadrupole magnets located downstream of the accelerating structures (DQ4, DQ5). We use a single-shot multislit diagnostic[25], also shown in Fig. 2, to measure the beam emittance. The principal challenge of conducting this measurement is that the beamline elements, while effecting the beam emittance, also modify the beam size and divergence at the diagnostic. Unfortunately, emittance measurements with this multislit emittance diagnostic have a finite dynamical range, limited to a narrow range of beam sizes and divergences. Such a limited dynamical range is a common problem shared by many types of diagnostics across accelerator facilities.

As a baseline for comparison we conducted a two-dimensional scan of the solenoid parameters (K0, K1). A 10 × 10 point grid, shown in Fig. 3, was created with upper and lower bounds determined from prior experimentation. For each set of

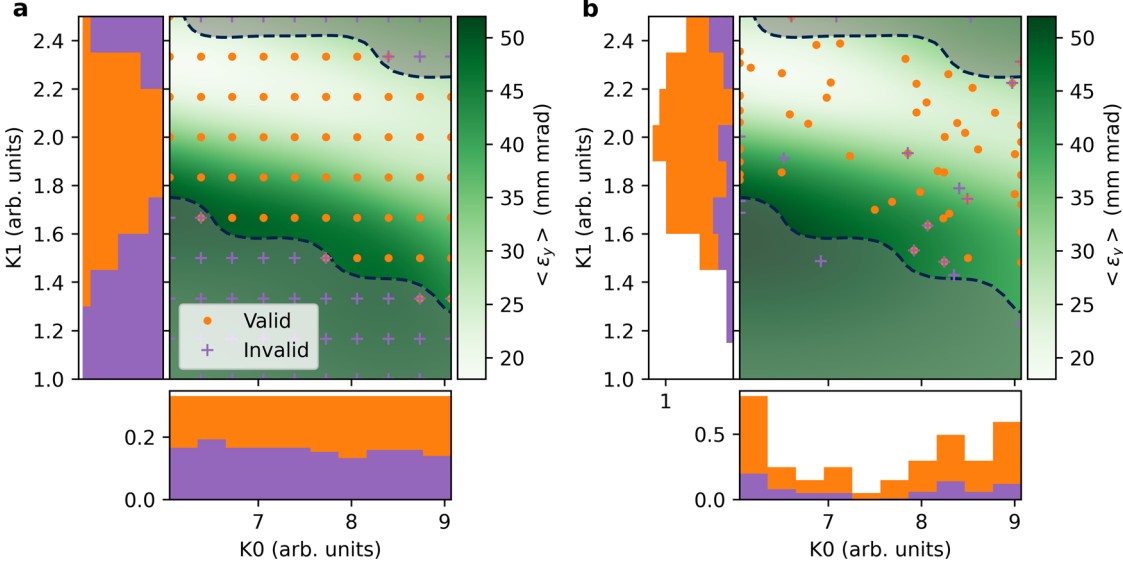

**Fig. 2 Cartoon depicting the emittance exploration experiment at the Argonne Wakefield Accelerator facility.** Conceptual view of emittance diagnostic is shown on the right. Beam travels from left to right.

**Fig. 3 Comparison between uniform grid sampling and CPBE. a** A 10 × 10 grid is sampled over an operational subdomain identified by prior experimentation. Color mesh represents the posterior predicted mean of a Gaussian process trained on the valid measurements. Stacked histograms show sampling density of measurements along each axis. **b** 2D projected samples from Bayesian exploration after 66 iterations with 4 parameters (K0, K1, DQ4, DQ5). In both plots, shading with dashed line denotes invalid region in subdomain with 50% confidence, calculated from uniform grid sampling.

parameters, five observations of the beam emittance were conducted while windowing the fluctuating bunch charge between 5.0 and 6.0 nC. In some cases, points in input space result in both valid and invalid measurements due to accelerator noise. We use these observations to train GP models of the vertical beam emittance $\varepsilon_y$ and a single constraining function $g(\mathbf{x})$ that models the measurement validity (see Methods Section), which is set to one for a successful measurements and zero otherwise.

From Fig. 3(a), we observe that the invalid region of input space (denoted by shaded regions), defined as $P[g(\mathbf{x}) \geq 0.5]$, is roughly half the input space domain. As a result, approximately half of our samples are wasted, as they provide no information on the beam emittance. We observe that in the valid region the emittance is strongly dependent on K1 relative to K0. If we normalize the input domain to the unit cube, the length scales of the GP kernel for each of these parameters is seen in Table 1. The length scales inferred from the data are consistent with what is seen in Fig. 3 where the emittance changes slowly as a function of the focusing solenoid strength (K0) and quickly with respect to the matching solenoid strength (K1).

We then used CPBE to explore a similar input space, varying the two solenoid magnet strengths (K0, K1) and two quadrupole-magnet strengths (DQ4, DQ5). We initialized the algorithm with two initial valid measurements, randomly generated from within

**Table 1 Trained hyperparameter length scales (normalized).**

| Parameter | Uniform grid sampling | CPBE |
|---|---|---|
| Focusing solenoid (K0) | 0.82 | 0.77 |
| Matching solenoid (K1) | 0.40 | 0.30 |
| Drive Quadrupole 4 (DQ4) | N/A | 1.03 |
| Drive Quadrupole 5 (DQ5) | N/A | 1.40 |

Kernel length scales of GP models trained on experimental data collected by both uniform grid sampling and Constrained Bayesian Exploration, normalized to unit cube input space.

the valid subdomain, determined from earlier experimentation. The sigma matrix for the proximal factor (given in normalized coordinate space) is set to $\mathbf{\Sigma} = 0.01\mathbf{I}$ where $\mathbf{I}$ is the identity matrix. This was chosen to reduce the acquisition function by $1/e$ over 10% of the input domain, which was identified to work well in practice. As in the 2D uniform scan, we use a single constraining inequality $g(\mathbf{x}) \geq 0.5$ where the constraint function $g(\mathbf{x}) = 1$ if the emittance measurement is valid and zero otherwise.

The results from this exploration, projected onto the 2D subspace where DQ4 = 0, DQ5 = 0 (focusing magnets are off), appear in Fig. 3(b). We again plot the posterior predictive mean of the emittance GP model in this subspace, trained on the sampled data. Estimated length scales from this model are shown

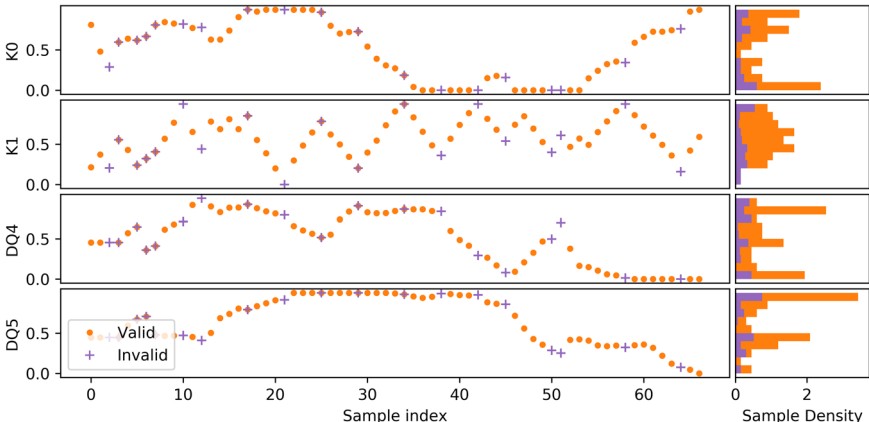

**Fig. 4 Tracing exploration path through input parameter space.** Normalized parameter values as a function of sample index during Bayesian exploration for all four input parameters. Corresponding sample density histograms are shown on the right.

in Table 1. Similar to the uniform grid model, the length scale for K0 is significantly longer than the length scale for K1, implying that the function varies faster along K1, which is consistent with what is observed in Fig. 3. Further, we observe that the length scales for K0, K1 are similar when comparing the two models. The difference between length scales for K1 is slightly larger, which could be explained by the larger average separation between points during grid sampling, relative to CPBE sampling. Grid sampling places limits on the fitted length scale, where the spacing of uniform samples determines the fastest changes in the target function that can be observed. Finally, the length scales for variables DQ4, DQ5 are larger than the normalized domain, implying that the emittance is weakly correlated with quadrupole-magnet strength, as expected by first order beam dynamics[26]. We even observe that the length scale for DQ5 is significantly larger than the length scale associated with DQ4, which possibly results from the magnet's location in the beamline, closer to the emittance diagnostic, that further reduces the impact of potential higher order effects from the magnet on the emittance.

From the projected histograms of K0 samples, we see that several regions of parameter space are avoided by the algorithm. This is due to the long length scale associated with K0, which reduces model uncertainty in between previous measurements. The algorithm skips over sampling these regions, as they do not significantly reduce uncertainty of our model if observed. On the other hand, samples along the K1 axis, which has a much shorter length scale, are continuously distributed in the valid region. This results in a better characterization the functional dependence on K1 since it shows rapidly changing behavior.

As a result, our sampling algorithm in 4D space significantly speeds up target function characterization over a hypothetical grid parameter scan. If we were to use the grid scan sampling algorithm in the 4D case, the number of samples needed grows exponentially up to $10^4$ samples. However, after only 66 samples taken by CPBE, we were able to produce a qualitatively similar model of the target function when compared to the previous 2D scan, representing a 150× speed increase. Furthermore, a larger portion of samples taken by the exploration algorithm are valid compared to uniform grid sampling (77% vs. 52%), providing more information about the emittance per sample on average. Note that invalid samples which appear inside the valid region shown in Fig. 3(b) are invalid due to correlations with the unplotted variables DQ4, DQ5.

The difference in sampling behavior is also observed in Fig. 4 where we plot the normalized trace of each parameter value during exploration. As a result of the short length scale associated with K1, the algorithm effectively scans back and forth across the

valid input region for this parameter while other parameters are slowly modified. The maximum and minimum values of the scan are dynamically determined when the algorithm encounters an invalid region. This is important, as we clearly see that the valid region for K1 changes as a function of the other variables. Finally, we observe that for variables with long length scales (K0, DQ4, DQ5), CPBE ignores intermediate regions that do not significantly reduce model uncertainty, reducing redundant measurements as it learns the associated length scale of the target function along these axes.

## Discussion

Here we have described and demonstrated an algorithm for autonomously and efficiently exploring input parameter spaces with limited prior information, while adapting to measurements that are potentially invalid or unsuccessful. This is achieved without the need for prior information about the target function or measurement constraints. We experimentally demonstrated our algorithm's ability to efficiently explore the functional dependence of the beam emittance on four accelerator parameters automatically, while navigating a practically difficult to execute measurement. The Bayesian exploration algorithm was able to achieve similar model prediction accuracy as a grid-based scan, with a significantly smaller number of samples and two extra input parameters. The advantages this algorithm confers over traditional methods of characterizing particle accelerator responses are substantial, and we expect it to have broad impact across accelerator-based science facilities. While we demonstrated this algorithm's effectiveness in the context of particle accelerators, CPBE is a flexible, lightweight, turn-key algorithm that replaces the need for grid type parameter scans in any field. In particular, the addition of a proximal biasing factor in our sampling strategy has advantages over previously described active learning algorithms when exploring spatially dependent systems (robotics, surveying etc.) that encounter costs associated with travel through input space.

While our demonstration here involved a relatively low-dimensional input space, there are a variety of methods that can be used to scale this algorithm to higher-dimensional parameter spaces. The main challenge facing this method is the computational cost associated with training and making predictions using the Bayesian model when the number of training points is large[17], as is often the case when operating in high dimensional input spaces. While the exact inference implementation of our algorithm is limited to lower dimensional spaces, several methods have been proposed to overcome this limitation. Of particular interest is the LOVE algorithm[27], which can compute covariances

used in our algorithm up to 2000 times faster than existing methods, without sacrificing accuracy. If approximate modeling can be tolerated, stochastic methods can be used with variational models that use a limited number of inducing points to approximate the posterior model distribution[28]. Finally, if samples can be evaluated in parallel, as in simulated exploration contexts, batched style methods could be used to generate batches of samples at each optimization step, similar to the acquisition functions implemented in[29].

## Methods

**Beam emittance diagnostic.** The geometric emittance of a beam can be determined experimentally using a multislit emittance diagnostic. This diagnostic consists of a transverse mask with a number of horizontal slits that divides the beam into multiple "beamlets". A downstream transverse diagnostic screen is used to image the beamlets and the center of mass, size, and integrated intensity of each beamlet is measured. This information is used to calculate aspects of the beam envelope, and thus the geometric beam emittance. To simplify our analysis, we used a slight modification of the emittance formulas derived in ref. [25]. Instead of calculating the emittance using the correlated divergence, we used a calculation of the uncorrelated divergence to determine the emittance. While this analysis prevents us from assessing the position-divergence correlation of the beam, it still provides us with an accurate emittance measurement.

Our experiment used a laser etched stainless steel mask with 25 slits. The slit pattern had a separation of 2 mm, a vertical slit width of 50 μm, and a horizontal slit length of 40 mm. The circular beam imaging screen, which had a diameter of 50 mm, was located 2.84 m downstream of the transverse mask. The screen was imaged using an optical camera with a spatial resolution of 46 μm per pixel.

Emittance measurements were only considered valid if the measurement satisfied three conditions: we required that (i) at least five beamlets were produced at the observation screen, (ii) all the beamlets were contained within a predefined region of interest on the screen to prevent biasing due to potential clipping, and (iii) the projection of each beamlet onto the vertical axis did not overlap with any other beamlet projections, in order to properly measure the size of each beamlet. If any of these requirements were not met, we assigned the constraining function at that location in input space a value of zero, tagging the measurement as "invalid".

**Gaussian process model creation.** Nonparametric Gaussian process surrogate models are used to predict the value of a target function $f(\mathbf{x})$ using Bayesian statistics[17]. These models are specified by a covariance function, $k(\mathbf{x}, \mathbf{x}'; \phi)$ with hyperparameters $\phi$ and a constant mean function $C$, such that we can write $f(\mathbf{x}) \sim \mathcal{GP}(C, k(\mathbf{x}, \mathbf{x}'))$. In an experimental setting, an observation $y$ is the target function corrupted by noise: $y = f(\mathbf{x}) + \epsilon$ where we assume that $\epsilon \sim \mathcal{N}(0, \sigma_{\text{noise}}^2)$. Given $N$ previous measurements $\mathcal{D} = \{(\mathbf{x}_1, y_1), \ldots, (\mathbf{x}_N, y_N)\}$ the predictive probability distribution of the function value $f = f(\mathbf{x})$ is given by

$$p(f|\mathcal{D}, \mathbf{x}) = \mathcal{N}(\mu(\mathbf{x}), \sigma^2(\mathbf{x})) \tag{5}$$

where

$$\mu(\mathbf{x}) = \mathbf{k}^T[K + \sigma_{\text{noise}}^2 I]^{-1}(\mathbf{y} - C) + C \tag{6}$$

$$\sigma(\mathbf{x}) = k(\mathbf{x}, \mathbf{x}) - \mathbf{k}^T[K + \sigma_{\text{noise}}^2 I]^{-1}\mathbf{k} \tag{7}$$

$$\mathbf{k} = [k(\mathbf{x}, \mathbf{x}_0), \ldots, k(\mathbf{x}, \mathbf{x}_N)]^T \tag{8}$$

$$K = \begin{bmatrix} k(\mathbf{x}_1, \mathbf{x}_1) & \cdots & k(\mathbf{x}_1, \mathbf{x}_N) \\ \vdots & \ddots & \vdots \\ k(\mathbf{x}_N, \mathbf{x}_1) & \cdots & k(\mathbf{x}_N, \mathbf{x}_N) \end{bmatrix}. \tag{9}$$

The model hyperparameters $\boldsymbol{\theta} = \{\phi, \sigma_{\text{noise}}, C\}$ are determined by maximizing the log marginal likelihood, $\boldsymbol{\theta} = \text{argmax}_{\boldsymbol{\theta}} \log[p(D; \boldsymbol{\theta})]$, which balances model accuracy and complexity when choosing hyperparameters. A Matérn kernel ($\nu = 3/2$) was used for the GP models in this paper. During experimentation the hyperparameters were retrained after each observation.

## Data availability

The observation data generated in this study have been deposited in the public repository https://github.com/roussel-ryan/turn_key_bayesian_exploration[30].

## Code availability

The code for our algorithm is located, along with code used to generate plots in this manuscript, in the github repository https://github.com/roussel-ryan/turn_key_bayesian_exploration[30].

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

## Acknowledgements

We would like to thank Sergei Nagaitsev for their discussions regarding the multislit diagnostic. This work was supported by the U.S. National Science Foundation under Award No. PHY-1549132, the Center for Bright Beams. The Argonne Wakefield Accelerator is supported by the U.S. Department of Energy, Office of High Energy Physics, under Contract No. DE-AC02-06CH11357.

## Author contributions

R.R., J.P.G.A., and A.E. wrote the manuscript; R.R. and J.P.G.A. designed the experiment; R.R. and W.L. wrote the software required to run the experiment; R.R., J.P.G.A., E.W., and P.P. ran the experiment; J.P., P.P., A.H., A.E., and Y.K.K. provided guidance and oversight, all authors contributed to the completion of the manuscript.

## Competing interests

The authors declare no competing interests.
