## [Peer Review File · Nature Communications]

Turn-Key Constrained Parameter Space Exploration for Particle Accelerators Using Bayesian Active LearningREVIEWER COMMENTS

Reviewer #1 (Remarks to the Author):

The paper is well written, it is possible to understand the method in sufficient detail, and I view the paper as an important contribution in the rapidly emerging study of automatic experimental design through Bayesian optimization schemes. The main novelty of the approach lies in the combination of constrained search with a proximity factor, both of which improves the applicability of Bayesian optimization methods to practical experiments, where such constraints on measurements and control parameters are common.

I recommend the paper for publication, after addressing the comments below.

1.Line 46: “samples a quasi-uniform grid”. I do not understand this: I had the impression that the search is not constrained to particular parameter values but was free. Please explain what you mean with this, or take it away if not applicable. Is it an instrumental limitation?

2.The base acquisition function chosen is the posterior uncertainty, which is optimal for maximizing information gain about the underlying function. But in this case, I understand the goal of the shaping of the beam to be a maximization of the emittance, in which case it would seem more natural to use the PI or EI acquisition functions as base, since maximizing the information about the underlying function (for example in regions of low emittance) is not the goal per se. Please motivate this particular choice with relation to the goal of the beam shaping.

3.The constraint function(s) $g_i(x)$ in eq 2: For this paper a single constraint function $g(x)$ was used as described in the methods section V.A? I would mention somewhere in the main text that for this application a single constraint function was used, if that is the case.

4.I would like to see a short motivation for the choice of the variance of the proximity function (chosen as 0.01 in line 136). I assume this is informed by the practice of handling the accelerator, but would like to see some motivation since it is this value in relation to $\sigma(x)$, the posterior of f at x , that determines whether e.g. bigger jumps are allowed, and so presumably determines to a large degree the efficiency of the algorithm.

5. The $2/3$ reduction in the number of samples is in comparison to gridding two parameters: If the grid scan had been done in all 4 dimensions, like the Bayesian exploration, the reduction would have been quadratically lower even if the length scales of those extra parameters are large, since this can not be assumed a priori for the grid search. Is there some reason that this is not emphasized more? It thus seems to me that the $2/3$ is a major understatement for the reduction in number of samples between a standard grid search and this method.

6. Some comments on syntax:

Line 85: "term" is usually for operands in addition, I would use "factor" for multiplication.

Line 113: "Too first order", should be "To .."

"dependant" everywhere: I think it should be "dependent", as in independent, dependency etc.

Best regards,

Jakob Svensson

Reviewer #2 (Remarks to the Author):

The main result of the presented study is successful experimental demonstration of a novel adaptation of the Bayesian optimization algorithm. The presented method mitigates the limitations of existing techniques used for characterization of beam properties which change in response to accelerator parameters. The biggest advantage of this method is the possibility to replace simple parameter scans, despite limited or missing prior information about the target function or constraints. The mathematical background of the new developed method is described in detail, providing the description of single terms and their explanation.

The paper presents an original extension of a popular optimization technique, the applicability of the method is demonstrated on a typical accelerator task. However, since such problems as characterization

of complex, non-linear systems with high-dimensional input parameter space, unrealistic simulations and measurements noise are common for various scientific domains, this work can be of great interest for other fields, but primary for general application in particle accelerators.

The results of experimental demonstration of the developed algorithm provide sufficient evidence for successful application in accelerators. Possible limitation of this demonstration is the relatively low number of input parameters and target variables (4 varying accelerator parameters and 1 objective), it would be interesting to explore how the algorithm scales with increasing the dimensions of parameter space or with additional/ combined measurements information on other beam properties.

The presentation of results is very well structured and easy to follow for the readers. The experimental set up and methodology are explained in sufficient detail, in a clear manner. Since the new method is compared to a baseline which is based on a typical accelerator scenario (emittance measurements), the benefits to the accelerator community are convincingly demonstrated.

I recommend the manuscript for publication, after considering following minor comments on the contents:

Line 113: “Too first order, [...]”, please correct to “To”

Line 116: [...], while effecting the the beam emittance”, please remove “the”.

Figure 3: It is difficult to see the shaded area in the lower part of the plots, maybe a dashed line to frame the area would be helpful.

References: please check the formatting of the reference [4]. Reference [26] does not provide the source, please add a link / publication if possible.

Reviewer #1

The paper is well written, it is possible to understand the method in sufficient detail, and I view the paper as an important contribution in the rapidly emerging study of automatic experimental design through Bayesian optimization schemes. The main novelty of the approach lies in the combination of constrained search with a proximity factor, both of which improves the applicability of Bayesian optimization methods to practical experiments, where such constraints on measurements and control parameters are common.

I recommend the paper for publication, after addressing the comments below.

1.Line 46: “samples a quasi-uniform grid”. I do not understand this: I had the impression that the search is not constrained to particular parameter values but was free. Please explain what you mean with this, or take it away if not applicable. Is it an instrumental limitation?

Bayesian Exploration samples points that are as far away from all other points as possible to maximize the information gain of each sample. If we initialize the algorithm with a set of regularly spaced observations in input space, the resulting sampling pattern resembles a grid with spacing dictated by the functional length scale associated with each axis (as shown in Figure 1). We used the nomenclature “quasi-uniform grid” to describe this behavior and to illustrate to the reader what the algorithm attempts to do. However, to avoid any possible confusion for the reader we modified the wording as follows.

Line 46: REPLACED “the algorithm sequentially samples a quasi-uniform grid of points in input space” WITH “ the algorithm sequentially samples points in input space that maximize information gain of the target function at each observation step.”

2.The base acquisition function chosen is the posterior uncertainty, which is optimal for maximizing information gain about the underlying function. But in this case, I understand the goal of the shaping of the beam to be a maximization of the emittance, in which case it would seem more natural to use the PI or EI acquisition functions as base, since maximizing the information about the underlying function(for example in regions of low emittance) is not the goal per se. Please motivate this particular choice with relation to the goal of the beam shaping.

The main goal of our algorithm, as is stated in the text, is to solve the characterization problem, not an optimization one. While a common goal of accelerator operation and optimization is emittance minimization, it is also often important to efficiently characterize the functional dependency of this beam property on tuning parameters. We

note in the text that characterization is a necessary part of accelerator operations, experimental planning and tolerance determination. To clarify this point in the text we made the following modification

Line 38: REPLACED “Our goal is to produce an algorithm that replaces simple parametric scans for function characterization, while meeting potentially poorly understood requirements for practical measurements in a flexible manner. The algorithm should be “turn-key”, requiring as little prior information and oversight as possible. An ideal algorithm will adapt its sampling strategy to the observed functional behavior in order to increase sampling efficiency. It should also determine the regions of input space which yield unsuccessful measurements and use this information to avoid invalid measurements in the future. This algorithm reduces beamline tuning time during normal operations, while enabling efficient exploration of novel or poorly characterized systems.” WITH “Our goal is to construct an algorithm that replaces simple parametric scans for characterizing a target function of interest, while meeting poorly understood requirements for practical measurements in a flexible manner. The algorithm should be “turn-key”, requiring as little prior information about the target function and oversight as possible. As a result, our algorithm reduces beamline tuning time during normal operations, while enabling efficient exploration of novel or poorly characterized systems.”

As you correctly noted, the base acquisition function that we use maximizes the information gain about the underlying function. We believe that this fact, along with the points described above (and highlighted in the text), properly motivates the use of posterior uncertainty as the base acquisition function. However, we did take this opportunity to further motivate the proximal optimization behavior of our algorithm.

Line 37: ADDED “Finally, it is desirable to prevent rapid changes in accelerator input parameters during operation. In some cases, it is temporally expensive to make changes in parameters, such as when mechanical actuators are used to change the phase of accelerating cavities. In other cases, fast feedback algorithms used in accelerator subsystems rely on adiabatic changes in external parameters to maintain system stability. Large jumps in parameter space can delay convergence of these feedback systems to stability or worse, cause them to fail entirely. Practical experimental considerations such as these must be considered when automated sampling algorithms are used, striking a balance between costs associated with changing input parameters and the information gained. “

3. The constraint function(s) $g_i(x)$ in eq 2: For this paper a single constraint function $g(x)$ was used as described in the methods section V.A? I would mention somewhere in the main text that for this application a single constraint function was used, if that is the case.

We have highlighted this in the text.

Line 123: REPLACED “and the measurement validity constraining function $g(x)$,” WITH “and a single constraining function $g(x)$ that models the measurement validity (see Section \ref{sec:methods},”

Line 136: REPLACED “the constraint inequality” WITH “a single constraining inequality”

4. I would like to see a short motivation for the choice of the variance of the proximity function (chosen as 0.01 in line 136). I assume this is informed by the practice of handling the accelerator, but would like to see some motivation since it is this value in relation to $\sigma(x)$, the posterior of f at x , that determines whether e.g. bigger jumps are allowed, and so presumably determines to a large degree the efficiency of the algorithm.

This is indeed the case. We added a small discussion in the text to motivate our choice for the proximity function.

Line 135: ADDED “This was chosen to reduce the acquisition function by $1/e$ over 10% of the input domain, which was identified to work well in practice.”

5. The $2/3$ reduction in the number of samples is in comparison to gridding two parameters: If the grid scan had been done in all 4 dimensions, like the Bayesian exploration, the reduction would have been quadratically lower even if the length scales of those extra parameters are large, since this can not be assumed a priori for the grid search. Is there some reason that this is not emphasized more? It thus seems to me that the $2/3$ is a major understatement for the reduction in number of samples between a standard grid search and this method.

This is an excellent point, we have made changes to the abstract and text to highlight this improvement.

ABSTRACT: REPLACED “We experimentally demonstrate that our algorithm autonomously conducts an adaptive, multi-parameter exploration of input parameter space, while navigating a highly constrained, single-shot beam phase-space measurement.” WITH “We experimentally demonstrate that our algorithm autonomously conducts an adaptive, multi-parameter exploration of input parameter space, potentially orders of magnitude faster than conventional grid-like parameter scans, while making highly constrained, single-shot beam phase-space measurements and accounting for costs associated with changing input parameters.”

Line 152, REPLACED “The mean emittance of the Gaussian process model trained from exploration samples is also qualitatively similar to the model generated by grid sampling, despite the exploration algorithm using roughly $2/3$ the number of samples.

” WITH “As a result, our sampling algorithm in 4D space significantly speeds up target function characterization over a hypothetical grid parameter scan. If we were to repeat the grid scan sampling algorithm used in the 2D case, the number of samples needed grows exponentially up to 10^4 samples. However, after only 66 samples taken by CPBE, we were able to produce a qualitatively similar model of the target function when compared to the previous 2D scan, representing a 150x speed increase.” **NOTE:** To improve readability we switched the order of the paragraphs in this area.

6. Some comments on syntax:

Line 85: “term” is usually for operands in addition, I would use “factor” for multiplication.

Line 113: “Too first order”, should be “To ..”

“dependant” everywhere: I think it should be “dependent”, as in independent, dependency etc.

Various syntax and readability changes were made in the text to address these and other issues.

Best regards,
Jakob Svensson

Reviewer #2

The main result of the presented study is successful experimental demonstration of a novel adaptation of the Bayesian optimization algorithm. The presented method mitigates the limitations of existing techniques used for characterization of beam properties which change in response to accelerator parameters. The biggest advantage of this method is the possibility to replace simple parameter scans, despite limited or missing prior information about the target function or constraints. The mathematical background of the new developed method is described in detail, providing the description of single terms and their explanation.

The paper presents an original extension of a popular optimization technique, the applicability of the method is demonstrated on a typical accelerator task. However, since such problems as characterization of complex, non-linear systems with high-dimensional input parameter space, unrealistic simulations and measurements noise are common for various scientific domains, this work can be of great interest for other fields, but primary for general application in particle accelerators.

The results of experimental demonstration of the developed algorithm provide sufficient evidence for successful application in accelerators. Possible limitation of this demonstration is the relatively low number of input parameters and target variables (4 varying accelerator parameters and 1 objective), it would be interesting to explore how the algorithm scales with increasing the dimensions of parameter space or with additional/ combined measurements information on other beam properties.

This is indeed an interesting avenue for future investigation. There are several potential options for expanding the applicability of our algorithm to larger input parameter spaces that are currently under investigation. Now that our initial algorithm has been successfully demonstrated in this manuscript we can proceed with more advanced research. We have added a discussion of current limitations of our algorithm and potential solutions to the discussion section of our manuscript.

Line 194 ADDED “While our demonstration here involved a relatively low-dimensional input space, there are a variety of methods that can be used to scale this algorithm to higher-dimensional parameter spaces. The main challenge facing this method is the computational cost associated with training and making predictions using the Bayesian model when the number of training points is large [17], as is often the case when operating in high dimensional input spaces. While the exact inference implementation of our algorithm is limited to lower dimensional spaces, several methods have been proposed to overcome this limitation. Of particular interest is the LOVE algorithm [27] which can compute covariances used in our algorithm up to 2000 times faster than existing methods, without sacrificing accuracy. If approximate modeling can be tolerated, stochastic methods can be used with variational models that use a limited number of inducing points to approximate the posterior model distribution [28]. Finally, if samples can be evaluated in parallel as in simulated exploration contexts, batched style methods could be used to generate batches of samples at each optimization step, similar to the acquisition functions implemented in [29].”

The presentation of results is very well structured and easy to follow for the readers. The experimental set up and methodology are explained in sufficient detail, in a clear manner. Since the new method is compared to a baseline which is based on a typical accelerator scenario (emittance measurements), the benefits to the accelerator community are convincingly demonstrated.

I recommend the manuscript for publication, after considering following minor comments on the contents:

Line 113: “Too first order, [...]”, please correct to “To”

Line 116: [...], while effecting the the beam emittance”, please remove “the”.

Figure 3: It is difficult to see the shaded area in the lower part of the plots, maybe a dashed line to frame the area would be helpful.

References: please check the formatting of the reference [4]. Reference [26] does not provide the source, please add a link / publication if possible.

Corrections have been made where possible with respect to the reviewers comments. Formatting corrections for references were attempted, but we seek further guidance from Nat. Comm. editors on correct formatting.

REVIEWERS' COMMENTS

Reviewer #1 (Remarks to the Author):

All comments and questions in my review have been properly answered and I recommend the article for publication.

Reviewer #2 (Remarks to the Author):

The authors have implemented the suggested corrections, the revised version of the manuscript addresses the comments mentioned in the review. The missing discussion of current limitations of the proposed algorithm has been added, clarifications on some technical details of the algorithms, such as the choice of hyper-parameter has been added as well.

Considering the response of the authors and the changes made, I recommend the manuscript for publication in its current revised version.